# Monte-Carlo based sensitivity analysis of the RIM2D hydrodynamic model for the 2021 flood event in Western Germany

Shahin Khosh Bin Ghomash[1], Patricio Yeste[2], Heiko Apel[1], and Viet Dung Nguyen[1]

[1]Section Hydrology, GFZ German Research Centre for Geoscience, Potsdam (Germany)
[2]Institute of Environmental Science and Geography, University of Potsdam, Karl-Liebknecht-Straße 24–25, 14476 Potsdam, Germany

**Correspondence:** Shahin Khosh Bin Ghomash (shahin@gfz-potsdam.de)

**Abstract.** Hydrodynamic models are crucial for flood forecasts and early warnings, particularly in response to events such as the devastating floods in Germany's Ahr region in July 2021. However, several uncertainties can be present in these models stemming from various sources, such as model structure, parameters, and boundary conditions. In this study, we aim to address these uncertainties and enhance the existing RIM2D hydrodynamic model introduced by Apel et al. (2022) for the Ahr region. The goal is to fortify its robustness and reliability for inundation simulations in the area. For this, we employ a large number of Monte Carlo simulations, assessing the effects of various model elements, such as floodplain and channel roughness coefficients, as well as terrain resolution, on river dynamics and inundation.

Our findings emphasize the critical role of proper parameter assignment in attaining optimal simulation results. The results show that sensitivity to input factors varies depending on the used performance metrics and the predicted output. We demonstrate that for simulating flow formation and water level in the river channel, the roughness parameter of the river channel and the model's resolution are paramount. On the other hand, for simulating flood extent and the distribution of water depth across the domain, even coarser resolutions prove adequate and, due to their reduced computation time, might be better suited for early warning systems. Furthermore, our findings suggest that the differences observed between finer and coarser resolution models may stem from the varying representations of the river channel and buildings within the model. Ultimately, this work provides a guideline for the parameterization of RIM2D and similar physically-based fluvial models tailored to the Ahr region, offering valuable insights for future hydrodynamic modeling endeavors in the area.

## 1   Introduction

The changing climate leads to an increased occurrence and intensity of extreme weather events, which leads to more frequent and more intense floods. In order to estimate the impact of the floods, hydrodynamic models are becoming more important (Li et al., 2019; Wu et al., 2017). Hydrodynamic models are used for flood management planning and risk assessments (Merz et al., 2020), but the need for the use of hydrodynamic modelling in operational flood forecast system has been shown in recent flood events and is thus advocated for by both the scientific community and flood managers (Šakić Trogrlić et al., 2022; Apel et al., 2022).

The critical importance of these models was highlighted by the devastating floods in Germany's Ahr region in July 2021 (Schäfer et al., 2021). In the aftermath of this disaster, Apel et al. (2022) established and presented a flood inundation model designed for the region. This model was built upon the RIM2D hydrodynamic model, specifically customized for swiftly simulating inundations in the affected region. Although the RIM2D model, primarily parameterized based on the modeller's expertise and standard hydraulic roughness values, demonstrated commendable accuracy in capturing the inundation dynamics, there remains room for refining and enhancing its precision and dependability, as it is common in initial modeling efforts.

Various uncertainties in hydrodynamic modeling can arise from different sources, including the model structure, parameters, and boundary conditions. These uncertain elements can have a substantial impact on the reliability and accuracy of the model's performance (Caviedes-Voullième et al., 2020; Alipour et al., 2022). Model structure uncertainty pertains to the inherent imperfections in the underlying equations and numerical schemes of a hydrodynamic model when simulating the physical processes within a river system. The primary parameters influencing hydrodynamic models are the river bathymetry and surface roughness coefficients (Khanarmuei et al., 2020). The selection of roughness coefficient values for both floodplain and channel, along with the resolution and accuracy of the Digital Elevation Model (DEM), can substantially influence the results of the hydrodynamic model (Pappenberger et al., 2006; Dung et al., 2011; Khosh Bin Ghomash et al., 2019; Alipour et al., 2022). To fully address the uncertainties inherent in hydrodynamic modeling and to obtain best possible model results, a calibration and validation process is typically applied (Fabio et al., 2010). This requires a large number of model runs (Alipour et al., 2022), but also sufficient data for evaluating the model performance. With sufficient data at hand, a large obstacle for hydraulic model calibration are the typically long simulation runtimes, which hinders the evaluation of a large number of model runs with different parameterisation. With the advancement of 2D shallow Water solvers like RIM2D, utilizing High Performance Computing (HPC) facilities like scientific-grade Graphic Processor Units (GPU), the evaluation of a large number of simulation runs became feasible. This allows for thorough model calibration and enhancement of the model's predictive capabilities, as well as the identification of optimal model parameters. With this in mind, our research aims to re-evaluate and calibrate the existing RIM2D model for the Ahr river, making it more robust and reliable for inundation simulations for the region. This is possible because of the comparatively well documented flood event in 2021, for which sets of different data for model evaluation are available.

In this study, we conduct an Monte-Carlo analysis to evaluate the spatio-temporal performance of RIM2D across varying resolutions and manning roughness parameters. The Monte Carlo simulation emcompasses 3,000 Latin Hypercube samples (Iman and Conover, 1982) of varying roughness parameters for different land use classes, and different spatial resolutions. To evaluate the performance of RIM2D, we utilize post-event inundation maps, observed water levels throughout the study area, and reconstructed water level hydrographs at gauging stations.

Firstly we introduce the study area and the datasets utilized. Subsequent sections will elaborate our research methods, the results obtained, and an in-depth discussion of these findings. The paper will conclude by exploring the implications of our work and its potential applications for future hydrodynamic modeling endeavors.

## 2    Case Study

The Ahr River, an 86 km long tributary of the Rhine River, is situated in the federal states of Rhineland-Palatinate and North Rhine-Westphalia of Germany. Our study area (Figure 1) concentrates on the downstream segment of the Ahr River, spanning about 30 km from the towns of Altenahr to Sinzig, where the valley remains relatively enclosed in the first third of the reach, after which the valley widens and extends downstream to the Rhine.

This area predominantly comprises rural landscapes, with a few small settlements and the comparatively larger urban area of Bad Neuenahr-Ahrweiler, with a population of approximately 26,500 (Truedinger et al., 2023). The region's average annual precipitation level is below the German average, measuring around 675 mm (Truedinger et al., 2023).

In mid-July 2021, heavy rainfall events in Western and Central Europe caused severe and abrupt flooding, particularly impacting Belgium, the Netherlands, and Germany (Schäfer et al., 2021). The Ahr Valley was among the severely affected areas, accounting for an overall 70 percent of all fatalities in the country (Truedinger et al., 2023). Various factors contributed to this extreme impact. The Ahr Valley, characterized by steep slopes and narrow valleys in a low mountain terrain, has been extensively settled and cultivated over an extended period in history, leading to a concentration of both population and structures in vulnerable zones. Consequently, such areas are inherently prone to hazards like rapid flow concentration, mass movement, erosion, and substantial debris accumulation.

During the flood event on July 14, 2021, water levels in the Ahr reached their highest values at the existing gauging stations since the beginning of their measurements reaching about 10 m water depth at gauge Altenahr (Mohr et al., 2022), although the exact water levels remain unknown due to damage or destruction of most gauging stations during the event (Schäfer et al., 2021).

## 3    Methods

### 3.1    Hydrodynamic model RIM2D

RIM2D (Rapid Inundation Model 2D) is a 2D raster-based hydrodynamic model developed by the Section Hydrology at the German Research Centre for Geoscience (GFZ) in Potsdam, Germany. Employing the local inertia approximation to the shallow water equations (Bates et al., 2010), a method well-established for fluvial floodplain inundation applications (Falter et al., 2014; Neal et al., 2011; Apel et al., 2022), RIM2D offers a robust computational framework.

In essence, the local inertia formulation provides a more accurate description of flood dynamics compared to more simplified versions of the shallow water equations, such as the diffusive wave model (De Almeida and Bates, 2013; Caviedes-Voullième et al., 2020). This approach introduces an additional term to represent the rate of change of local fluid momentum, influencing the progression of fluid momentum from one time step to the next. In practical terms, this implies that the fluid's momentum in a given time step informs the subsequent step, necessitating an acceleration of the flow from its previous state. Consequently, in describing physically shallow water flows, the local inertial formulation bridges the gap between the diffusion wave approximation and the comprehensive full-dynamic equations.

While the original numerical solution proposed by (Bates et al., 2010) is susceptible to instabilities under near-critical to super-critical flow conditions and for small grid cell sizes (De Almeida and Bates, 2013), RIM2D incorporates the numerical diffusion method suggested by De Almeida et al. (2012) to address these issues.

RIM2D is coded in Fortran and ported to GPUs using NVIDIA CUDA Fortran libraries, enabling massive parallelization of numerical computations at a lower cost compared to large multi-core computing clusters. It's important to note that, at present, RIM2D exclusively supports computations on a single GPU, which limits the amount of parallel computations to about 1-2 millions depending on the GPU type. However, a multi-GPU version will available in the near future.

## 3.2 Data and model setup

The model setup utilized the DGM1 product, a digital elevation model (DEM) with a resolution of dx = 1 meter, provided by the Landesamt für Umwelt (LfU) of the Federal State of Rhineland-Paltinate. DGM1 was created through LiDAR (Light Detection and Ranging) mapping of the region in 2016. In this study, simulations were conducted using two resolutions, namely dx = 5 meters (5.6 million cells) and 10 meters (1.4 million cells). Each DEM employed in the simulations resulted from aggregating the DGM1 product through the averaging method. The resulting DEMs were directly used as the foundation for the simulations without undergoing any additional alterations. Consequently, the simulations may not accurately represent the riverbed but instead depict the average water surface in the Ahr river, typically measuring less than 1 meter (Apel et al., 2022). This simplification is suitable for the model's goal of simulating extreme flows that significantly exceed average flow depths but also justified as RIM2D operates based on water levels as boundary conditions rather than water depths and discharge. Thus, even with the presumed bed elevation, the water levels at the model boundary consistently remain accurate, ensuring over-bank flow and floodplain inundation occur in the correct locations and at the appropriate times. The advantage of this simplification is that it allows the model approach to be applied to any river reach without requiring detailed local knowledge of river bathymetry. This facilitates an easy, semi-automated, and cost-effective implementation and transfer of the model approach to virtually any location where a DEM with adequate resolution relative to the river width is available. A free outflow condition is applied to all boundaries of the simulation domain, allowing water from the Ahr River to flow into the River Rhine and exit through the domain boundaries. To prevent simulations from starting with empty river conditions, an initial water depth of 1 meter, reflecting the Ahr River's typical depth, is assigned. Additionally, an 18-hour warm-up period before the onset of the flood using low flow boundary data from the available Altenahr gauge time series is applied in all simulations. Building footprints in the simulation domain were excluded from the two DEMs based on building shape files provided by OpenStreetMap. An example of this is shown in Figure 1 (white coloring in the lower figure). This means that buildings act as closed reflective boundaries for the flow in the simulations, just as in reality.

Each simulation was performed on a single Nvidia A100 GPU cluster, with computational times of approximately 28.8 minutes and 5.8 minutes for resolutions of dx = 5 meters and dx = 10 meters, respectively. The primary reason for not conducting this calibration exercise at higher resolutions was the high computational cost. At a resolution of dx = 1 meter, the domain contains 139 million cells, while at dx = 2 meters, it has 34.7 million cells. This results in runtimes of approximately 5.3 hours for dx = 2 meters and 25.9 hours for dx = 1 meter. These extensive runtimes made it impractical to run the large number of

simulations required for our calibration needs in this study. Furthermore, Khosh Bin Ghomash et al. (2024a) demonstrated that resolutions of dx = 5 and 10 meters are sufficient for accurately simulating flood extents in the Ahr Valley, capturing critical details of flood dynamics without the need for finer resolutions. This finding supports the use of these resolution levels for effective flood modeling, balancing computational efficiency with spatial resolution.

Manning roughness values were assigned to the domain based on the 2020 Germany land cover classification derived from Sentinel-2 data (Riembauer et al., 2021). The data basis for the classification includes atmospherically corrected Sentinel-2 satellite data (MAJA algorithm; data provided by EOC Geoservice of the German Aerospace Centre – DLR), training data from reference data (e.g., OpenStreetMap), and the Sentinel-2 scenes themselves. This land cover was chosen for its relatively high grid resolution (10 meters). Additionally, the main Ahr river channel in the domain was incorporated into the land cover raster as a new land category. The land cover is depicted in Figure 1.

Subsequently, the land cover is divided into seven main categories which are presented in Table 1. After an extensive literature review, a diverse set of Manning value ranges was selected for each land category based on (Te Chow, 1959; Arcement et al., 1989; Goodchild et al., 1993). The lower and upper boundaries of the Manning roughness values for each land category are detailed in Table 1. Additionally, the coverage percentages of each land category concerning the entire domain and the observed 2021 flood extent are presented in Table 1. The Manning values assigned to each land-cover are then varied during the Monte Carlo analysis as described in Section 3.3. Given their location mainly downstream near the Rhine River, water bodies are anticipated to have minimal impact on the model results. As a result, our analysis focuses on the impact of six categories: 'Forest', 'Vegetation', 'Built-up', 'Bare soil', 'Agriculture', and 'River channel'.

The water level data (in meters above sea level) at Altenahr and Bad-Bodendorf gauges, officially reconstructed by the LfU Rhineland-Palatinate, were employed in this study (Mohr et al., 2022). The time series reconstruction was necessary because the gauges were destroyed during the 2021 event. The reconstruction relied on hydraulic simulations, which has to be taken into account when assessing the model performance. The Altenahr time series was used as input forcing for the models and the Bad-Bodendorf gauge time-series was used for validation. For model setup, observed Altenahr water levels were assigned to the inflow cells in the domain, selected along the river channel on the west boundary of the domain. To account for over-bank flow into the domain, cells neighboring the river channel with elevations below the maximum water level of the flood hydrograph were also selected as boundary cells. Water depths were assigned to the chosen cells only when the river water levels exceeded the cell elevation. In this work, we analyze both the simulated flood extent and maximum water depths across the entire domain and within the river channel, comparing them to observed data. For this, we employ the documented flood extent in 2021 (supplied by the German State Agency for the Environment, LfU - Landesamt für Umwelt) and water depths derived from 65 high water marks (Figure 1) reported by residents (Apel et al., 2022), facilitating the evaluation of simulated water depths.

### 3.3 Monte-Carlo based sensitivity analysis

A Monte-Carlo based sensitivity analysis over the roughness parameterization was conducted in order to explore the spatio-temporal performance of RIM2D at 5m and 10m resolution and test its predictive capability for the flood event in the Ahr

valley in 2021. The uncertainty analysis was founded on a Monte Carlo simulation comprising 3,000 Latin Hypercube (LH) samples (Iman and Conover, 1982) of the roughness parameter space defined by the roughness ranges indicated in Table 1. The performance of RIM2D was assessed using observations available for the flood event under study according to the fol-
160 lowing procedure: 1) the in-channel flood dynamics were evaluated against the reconstructed water level time series at the Bad Bodendorf gauge, 2) the simulated inundation extent was compared with the post-event map of the inundated area, and 3) the simulated water depths were compared against the 65 high water marks at buildings reported by the inhabitants.

The performance of RIM2D for the in-channel flood dynamics was determined through the root-mean-square error (RMSE) and the Kling-Gupta efficiency (KGE, Gupta et al., 2009) applied to the water level at the river gauge. RMSE and KGE do not
present an upper and a lower limit, respectively, with RMSE = 0 and KGE = 1 indicating a perfect match. The performance for inundation extent was assessed through four metrics commonly used to evaluate hydrodynamic models (e.g., Bryant et al., 2024; Wing et al., 2017): the hit rate ($H$), the false alarm ratio ($F$), the critical success index ($C$) and the error bias ($E$). To compute these metrics, the maximum inundation maps of the simulations are evaluated against the observed flood extent of the event. These metrics are based on a binary comparison of wet and dry cells according to a classification into false positives
(i.e., wet cells in model corresponding to dry cells in observations), false negatives (i.e., dry cells in model corresponding to wet cells in observations), true positives (i.e., wet cells both in model and observations) and true negatives (i.e., dry cells both in model and observations). $H$ measures the tendency of the model to underpredict the inundation extent and ranges from 0 (no true positives) to 1 (no false negatives). $F$ indicates the tendency of the model to overpredict the inundation extent, ranging from 0 (no false positives) to 1 (no true positives). $C$ reports on both underprediction and overprediction and also ranges from
0 (no true positives) to 1 (no false positives and no false negatives, which corresponds to a perfect match between the simulated and the observed inundation extent). $C$ is the most commonly used performance criteria for comparing mapped and simulated inundation extent, because it combines $H$ and $F$. $E$ assesses the tendency of the model to underpredict or overpredict, with $E = 1$ indicating no bias, $0 \leq E < 1$ suggesting a tendency towards underprediction and $E > 1$ toward overprediction. Lastly, RMSE and the bias (i.e., mean observed water depth minus mean simulated water depth) were selected as the performance
metrics for the distributed water depths derived from the high water marks.

A final selection of the best-performing roughness sets representing the overall model performance was carried out according to a Euclidean distance (ED) definition applied to KGE, $C$ and RMSE for the distributed water depths after normalising the metrics between 0 and 1 as follows:

$$\text{ED} = \sqrt{(1 - \text{KGE}_\text{n})^2 + (1 - C)^2 + (1 - \text{RMSE}_\text{n})^2} \tag{1}$$

where $\text{KGE}_\text{n}$ and $\text{RMSE}_\text{n}$ are the normalised KGE and RMSE for the distributed water depths, respectively, as given by:

$$\text{KGE}_\text{n} = (2 - \text{KGE})^{-1} \tag{2}$$

$$\text{RMSE}_\text{n} = \left(1 + \frac{\text{RMSE}}{\bar{d}_{obs}}\right)^{-1} \tag{3}$$

where $\bar{d}_{obs}$ is the mean observed water depth for all the high water marks.

On the basis of Eq. 1 a total of 150 sets (5% of the 3000 sample parameter sets) corresponding the lowest ED to the ideal solution $(1, 1, 1)$ were selected and their corresponding parameter distributions were compared to evaluate the commonalities and differences between the dx = 10m and dx = 5m simulations in terms of multi-criteria model performance. The roughness ranges associated to the selected roughness sets can be considered as representative of the optimal RIM2D performance and provide generalizable recommendations for flood inundation studies using 2D raster-based hydrodynamic models.

## 4   Results and Discussion

### 4.1   Simulating water level at the Bad-Bodendorf gauge

Figure 2 illustrates the simulated water levels at the Bad-Bodendorf gauge station in simulations with spatial resolutions of dx = 10 m (a) and dx = 5 m (b). The black line represents the officially reconstructed water level during the 2021 flood event at the gauge, serving as a validation reference. In general, the simulated water levels exhibit a consistent overall shape with the validation curve. The peaks and recessions align perfectly, but differences emerge in the rising limbs.

Specifically, the dx = 5 m simulations demonstrate a better capture of the onset compared to the dx = 10 m simulations. The latter exhibits a delayed but also significantly steeper rising limb relative to both the validation curve and the dx = 5 m simulations. This suggests that simulation resolution directly influences the onset of the curves and the establishment of the flow. The delay and increased steepness in the rising limb of the simulated water level time series can be attributed to the properties of the underlying hydraulic equation and the properties of the numerical scheme, which tend to steepen the flood wave front and slower flood wave propagation in the shock-wave like flash floods as simulated with unsteady flow condition at the wave front (De Almeida and Bates, 2013). According to De Almeida and Bates (2013), while the local inertial approximation is computationally efficient, it can produce significant depth gradient errors as the Froude number nears unity. Such conditions are characterized by amplified depth oscillations, especially at higher Froude numbers, which may partly explain the rapid accumulation and abrupt arrival of the flow front observed in the simulated water level time series.

Moreover, coarser resolution models often fail to accurately capture the detailed topography and smaller features of a river channel, such as its banks and bends. Given that the Ahr river channel's width is at max around 10 m, a resolution of 10 m may not adequately represent the riverbanks, causing parts of the floodplain to "leak" into the river channel as represented in the 10 m DEM. This reduces the channel's conveyance capacity, leading to an oversimplification of water flow through the channel, which in turn affects the timing and shape of the hydrograph. Specifically, the delay in the rising limb is likely due to the model's inability to accurately represent the initial flow establishment in the river channel. When using a coarser mesh, the model may misrepresent how water fills the channel and interacts with surrounding floodplains, resulting in a slower onset of the simulated hydrograph compared to reality. This delay can create a more abrupt increase in water levels once flow is fully established, resulting in a steeper rising limb. In addition, the steepness may be influenced by the way the coarse model handles flow velocities and the transition from bankfull conditions to overbank flows. With finer resolutions, models can better simulate the gradual inundation of floodplains and the dispersion of water, leading to a smoother transition in the hydrograph.

However, with a coarse resolution, this transition tends to be more abrupt, making the simulated hydrograph steeper. In such situations where accurately simulating water levels within the river channel is crucial, and the available DEMs are too coarse to properly represent the channel, the use of 1D-2D modeling approaches that incorporate detailed river bathymetry or cross-section data may be beneficial. These 1D–2D hybrid approaches have been shown to perform well in hydrodynamic flood modeling (Echeverribar et al., 2020; Noh et al., 2018).

This observation of differences between the coarser and finer setups also aligns with findings from prior studies which also noted delays in runoff onset with coarser DEM resolutions under pluvial rainfall/runoff simulations (Caviedes-Voullième et al., 2012; Khosh Bin Ghomash et al., 2019).

To assess how the simulated water levels compare with the reconstructed water level time series at the Bad-Bodendorf gauge, Figure 3 illustrates the RMSE and KGE indicators for simulations with grid resolutions of dx = 10 m (a) and dx = 5 m (b). The metrics specific to each land cover class within the simulated area are depicted to analyze the influence of each class on the outcomes.

Examining the RMSE, it is evident that variations in the Manning values for forest and river channel classes have the most significant impact on model accuracy with both model resolutions. Lower Manning values for the river channel and higher Manning values for the forest land cover lead to the highest model accuracy in the dx =5 m setup. This pattern also holds true for the river land cover in the dx = 10 m simulation cases, but not for the forest land cover. Within the Forest class, noticeable differences exist between the dx = 10 m and dx = 5m setups. The dx = 10 m setup consistently leans towards smaller roughness values, approximately 0.1, signifying an increased sensitivity to variations in roughness. In contrast, the dx = 5m setup demonstrates an inclination for higher roughness values, typically around 0.25, indicating an alternative sensitivity pattern within this particular land use class. Interestingly, the model accuracy in simulating the water level at the Bad-Bodendorf gauge appears to be relatively unaffected by changes in the Manning values of other land cover categories. From the perspective of the KGE indicator, no dominant trend is observed among the Manning value ranges of different land covers, except for the river channel category in the dx = 5 m simulations. Here, lower Manning values result in a superior KGE value, aligning with the observations from the RMSE indicator. Our findings indicate that the Manning's value assigned to the river channel has a dominant impact on simulating water levels within the channel. This is consistent with previous research (Hall et al., 2005; Pappenberger et al., 2008; Bhola et al., 2019). For instance, Pappenberger et al. (2008) conducted a sensitivity analysis of a flood inundation model on the River Alzette in Luxembourg and showed that the channel roughness coefficient has a considerably greater impact on river water dynamics compared to floodplain roughness values.

Concerning resolution, it is evident that, overall, setups with a higher resolution (dx = 5m) achieve greater accuracy in simulating the water level at the Bad-Bodendorf gauge compared to the coarser setups with dx = 10m. This is reflected in their superior RMSE and KGE scores, as also seen in Figure 2. Notably, setups with higher resolutions demonstrate increased sensitivity to Manning roughness values, evident in relatively broader spreads in RMSE and KGE scores compared to setups with coarser dx = 10m. This emphasizes the significance of carefully selecting roughness values in such setups. According to our analysis, predicting water level dynamics is predominantly influenced by the resolution of the Digital Elevation Model (DEM). This aligns with prior research, such as Alipour et al. (2022), which identified DEM and mesh resolutions as the

most critical factors in simulating water level dynamics. Modelers have various methodologies at their disposal to improve the representation of small-scale features in coarser DEMs, thereby achieving the most accurate depiction of topography (Fewtrell et al., 2008). The significance of DEM resolution has also been underscored in other studies that explored the sensitivity of water level and inundation area to hydrodynamic model factors (Khosh Bin Ghomash et al., 2019; Oubennaceur et al., 2019).

## 4.2 Skill for the simulated inundation extent

Figure 4 depicts the model performance metrics, including Hit Rate (H), False Alarm ratio (F), Critical Success Index (C), and Error Bias (E), evaluating the efficacy of simulations at resolutions of dx = 10 (a) and 5 (b) meters in reproducing the inundation extent observed during the 2021 flood event. These metrics provide insights by comparing the maximum simulated flood extent from the simulations with the observed extent of the 2021 Ahr flood. Furthermore, the metrics are analyzed with respect to different land cover categories within the simulated region to discern the influence of each land type on the fidelity of simulated inundation areas.

Generally, it is evident that RIM2D demonstrates the ability to simulate flood extent with remarkable accuracy across various resolutions and nearly all roughness settings. This is shown by the high scores across nearly all metrics, indicating high performance. Of particular importance is the Critical Success Index (C), as it combines matches and mismatches in one metric. The obtained values cluster around C = 0.9, which is exceptionally high in simulations of flood extent. The most distinct contrast in the accuracy of the simulated flood extents between the two resolutions becomes evident through the Error Bias metric. An Error Bias value of 1 suggests an unbiased result, with scores closer to 0 indicating under-prediction. It becomes apparent that in the coarser dx = 10m configurations, the scores tend to be higher, approaching values closer to 1, while in the dx = 5m setups, there is a tendency towards underestimating the flooded regions compared to the coarser setup. This discrepancy in flooded area sizes observed between coarser and finer simulations might stem from artificially extended flooded regions with shallow depths in the larger cells of the coarser grids, contrasting with the smaller flood extents characterized by deeper water at the fringe of the inundation area in the finer grid configurations.

Regarding the influence of land classes, the H, F, and C metrics display limited sensitivity to changes in Manning values associated with each land category, with indicators generally remaining within the same range. However, the Error Bias indicator reveals a broader dispersion in results, indicating that changes in roughness Manning values may exert a more pronounced effect on the error bias of simulated flood extents across the domain. Overall, while no clear trend emerges regarding the impact of Manning roughness values on different land classes, the Error Bias indicator highlights the sensitivity of flood extent accuracy to such variations in the domain.

## 4.3 Comparison to water marks

Figure 5 illustrates the Root Mean Square Error and bias metrics, comparing simulated water depths to observed water marks during the 2021 flood event with grid spacing of dx = 10 m (a) and dx = 5 m (b). Each land cover class within the simulated area is analyzed separately.

In general, it is evident that the coarser dx = 10m setup tends to yield better scores across the two indicators in contrast to the finer setups when simulating water depths at various locations within the domain. This is primarily due to the worse scores observed at points farther from the river channel in the finer resolution setups. In these cases, because of the smaller simulated flood extent (as discussed in Figure 4), less water reaches these points, resulting in more significant discrepancies compared to the coarser dx = 10m simulations. Additionally, the finer resolution setups demonstrate a heightened sensitivity to changes in the Manning values, resulting in wider spreads in their scores, underscoring the importance of selecting appropriate roughness Manning values for these setups (also seen in Figure 3. These results underscore the suitability of dx = 10 m resolution for flood inundation modeling and early warning systems, providing sufficient accuracy for simulating water depths across the domain.

In respect to the Manning values assigned to different land classes, a slight trend is noticed in the river channel class across both resolutions. Lower roughness Manning values are associated with higher simulation accuracy, which is in line with the observations from Figure 3 regarding water levels at the Bad-Bodendorf gauge. In contrast, the Built-up class, comprising the largest proportion of the flooded area in 2021 (Table 1), shows a tendency toward higher Manning values in the dx = 10 m setups, while this trend is less evident in the finer dx = 5 m setup. This finding points towards less realistic representation of the built-up structure in the 10 m DEM resolution, in which buildings with footprints smaller than 10 m might disappear during rasterization of the OSM vector files. The impact of rasterization is evident in Figure 6, wherein the coarser DEM (a) fails to depict smaller buildings, typically smaller in length compared to the DEM resolution. Consequently, fewer obstacles obstruct the flow, attributable to the reduced level of detail in the representation of buildings in the coarser DEM. In order to compensate for the missing buildings as obstacles to the flow, higher roughness values are selected to to capture the mapped inundation depths in the urban areas. A similar, albeit less pronounced, trend is observable within the Forest class, where lower Manning values correspond to reduced errors. Conversely, Vegetation, Bare soil, and Agriculture classes appear relatively insensitive to changes in roughness Manning values, with no clear trend discernible.

## 4.4 Best overall performance

Figure 7a shows the parameter distribution of the 150 best-performing parameter sets according to the overall performance as estimated by the Euclidean distance (ED, see Eq. 1). ED is a comprehensive metric for simulating the in-channel flood dynamics, inundation extent, and distributed water depths across the domain. Figure 7b, c, d display the probability density of the top performers for KGE values of the simulated water level at the Bad-Bodendorf gauge, the C indicator for the simulated inundation extent, and the RMSE values for the simulated water depths compared to the observed water marks, respectively. It is important to mention that the values presented in the figure are based on the 150 best-performing Latin Hypercube (LH) samples for each individual performance metric and for each land use class. Detailed information regarding the optimal roughness ranges based on the best overall performance for every land class is also provided in Table 2.

In terms of best overall performance as indicated by ED and shown in Figure 7a, it is observed that both resolution setups tend to favor relatively similar ranges of Manning values for each land class. However, the coarser dx = 10m setup exhibits a more concentrated distribution and smaller spreads throughout the classes, except for the river channel class. Within the River

Channel class, the better performing dx = 5m simulations tend to focus on a narrower range of Manning values compared to the coarser setups. This highlights the increased importance of assigning appropriate Manning values to the river channel in finer resolution simulations. At coarser resolution, in which the channel is less realistically represented, the channel roughness is of lesser importance. Following the river channel, the Forests and Built-up classes (which are the predominant land use classes surrounding the river channel) display a narrower range of Manning values, underscoring their significant influence on the simulation outcomes. This provides experimental evidence for the theoretically reasonable importance of Manning roughness values for land use types that have substantial impact on the flood propagation, i.e. those classes predominant in the floodplains. In order to better understand the differences between both resolution setups, the RIM2D simulations corresponding to the lowest ED are shown in Figure 8. Consistently with our previous analysis, it can be appreciated that the dx = 5m setup performs better for the water levels at the Bad-Bodendorf and the inundation extent, whereas the dx = 10m setup achieved better performance for the distributed water depths.

Concerning the KGE indicator for the simulated in-channel flood dynamics (Figure 7b), there are relatively similar trends and ranges of Manning values observed compared to the ED indicator. Notable differences arise in the river channel class, where the coarser simulations exhibit a broader spread compared to the finer setups. This highlights the impact of inadequate representation of the river channel in these simulations, particularly for modeling water level dynamics (i.e. timing) along the course of the river channel. Additionally, discrepancies are evident in the preferred Manning values within the Forest class. The finer setups tend to favor higher Manning values around 0.25, whereas the coarser simulations lean more towards lower values. This observation aligns with the findings in Figure 3 as well.

The general patterns and Manning roughness ranges observed in the preceding two indicators are also consistent with the C values (Figure 7c). The sole deviation arises in the Vegetation category, where higher Manning values are favored to achieve improved inundation extents. This differs from the preferred Manning values used to simulate water levels at the Bad-Bodendorf gauge, which can be attributed to the significant presence of Vegetation class within the observed inundated areas and its distribution not solely along the river channel but at some distance in the inundated areas (see also Table 1 and Figure 1). Consequently, the Manning values assigned to this category play a more crucial role in simulating flood extents in the simulations. Another noteworthy observation from the figure is that the most effective Manning values for simulating water levels in the river may not always yield the most accurate simulated flood extent. This offers essential insight that when fine-tuning hydrodynamic models for flood inundation modeling, solely calibrating the model based on the water level or discharge time series at one point in the river—often a common practice—is inadequate. Such calibration methods do not necessarily lead to the most precise simulated flood extent. However, in this evaluation it has to be noted that the time series used for evaluating the in-river water level is also a simulation product. This means that the reconstruction is also prone the above mentioned errors and thus is likely not absolutely correct, which weakens the finding drawn from our experiments to some extent.

The RMSE values corresponding to the distributed observed water depths exhibit patterns and Manning ranges akin to those observed in the C indicator for the best inundation extents (Figure 7d). For instance, the Manning ranges favored for the Vegetation class align closely with the optimal C range, as opposed to the range for the preferred Manning values used in simulating water levels at the Bad-Bodendorf gauge. This underlines the significance of Manning values assigned to land-use

classes situated away from the river channel when modeling variables such as flood extent and water depths outside the river channel itself.

## 5 Conclusions

Decisions made by modelers during the setup phase significantly shape the outcome of the model, leading to dependable and efficient flood risk simulations. Understanding the model's behavior is crucial in gauging how the definition of its factors influences the desired output. For hydraulic models the model setup and the definition of the hydraulic roughness is the most important model parameter to influence the model results. This study employed 6000 different model setups and roughness parameterizations to assess the impact on the performance of the RIM2D Ahr valley hydrodynamic model, differentiated in several aspects. A variety of performance metrics were implemented to assess the quality of the simulated river water levels, flood extent, and distributed water depths in the simulation domain compared to the observed data from the 2021 Ahr flood.

With the comprehensive analysis, we observed that the sensitivity to different model resolutions and roughness combinations varies depending on the scale of sensitivity, the type of performance measure used, and the predicted output. For simulating the dynamics of water levels in the river channel, we found that the most crucial factors are the roughness parameters designated for both the river channel itself and the land classes bordering the channel, as well as the spatial resolution of the model setup. Specifically, the resolution plays a pivotal role in the initial stages of flow formation, where coarser resolutions result in a delay in the onset of runoff compared to finer resolutions. This delay can primarily be attributed to the inadequate representation of the river channel in the coarser setups, emphasizing the significance of considering this aspect when configuring models for simulating water levels in the river channel.

In terms of flood extent dynamics, which are arguably a more crucial aspect in flood inundation modeling and subsequent calculations of flood damages and risks, the findings of this study suggest that a coarser resolution DEM with dx = 10 m is perfectly adequate for generating satisfactory flood extent representations. This insight is particularly valuable for establishing early warning systems using such physically-based models, where the computational burden of simulations typically influences the choice of model parameters. Furthermore, it is demonstrated that, in the case of simulated flood extent, besides the Manning values of the river channel and its surrounding land categories, the Manning values assigned to the predominant land classes in the domain also significantly impact the accuracy of flood extent simulations. These assigned Manning values are crucial for achieving adequate flood extent simulations, a result also highlighted by Alipour et al. (2022), that also holds true for the simulation of distributed water depths throughout the domain. In our work, we show that the already well performing, but un-calibrated model of Apel et al. (2022) can be further improved using the identified best performing roughness parameter sets in this study. Moreover, we emphasize that for the RIM2D hydrodynamic model to be successfully implemented in other regions, it is crucial to take into account regional hydrological characteristics and adjust model parameters as needed. Important factors include understanding local topographic complexity and dominant land cover types.

The variation in sensitivity analysis results based on the specific model output highlights the complexity of such flood modeling. This shows that it is unlikely for a single factor to consistently be the most influential across all types of model

outputs. Our findings demonstrate that the sensitivity of RIM2D (and similar models) predictions is intricate, and modelers should carefully prioritize input factors according to the desired model output. Based on the results of this sensitivity analysis, modelers should select input factors that align with the desired model output and the specific application setup of the model. For simulations focused on river channel dynamics, and when computational cost is not a limiting factor, it is advisable to use finer resolutions, especially in areas with complex topography like narrow river channels, and to optimize the Manning roughness values for both the river channel and its bordering land classes. Conversely, when the primary focus is on flood extent or water depths within the floodplain, or when computational constraints are present, prioritizing the optimization of Manning roughness coefficients for the floodplain and employing coarser resolution DEMs is recommended. Implementing these strategies can lead to more accurate flood risk assessments, ultimately helping to mitigate and reduce the impacts of future flood events.

Additionally, an important conclusion from this study is that the optimal parameters for simulating water levels along the river channel may not necessarily align with the indicators necessary to achieve the best flood extent. In other words, calibrating such models based solely on water level or discharge time series at one point along the river is insufficient for optimizing a model for flood inundation calculation and subsequent damage and risk assessments.A potential approach to improve calibration is to incorporate spatio-temporally distributed water surface elevation datasets obtained from satellite missions such as SWOT and ICESat-2, as well as high-resolution data from drone surveys (Jiang et al., 2020; Bandini et al., 2020). These datasets can provide more detailed information across various sections of the river and floodplain, allowing for a better representation of the spatial dynamics of flooding. Incorporating such data could enhance the model's ability to simulate flood events more accurately, ultimately improving the reliability of flood risk and damage assessments.

Overall, this study provides distinct outlines and recommendations for configuring physically-based hydrodynamic models for fluvial floods in the Ahr area in Germany. It delineates the best-performing ranges of roughness Manning values for the domain (as detailed in Table 2) and offers guidance on selecting the model resolution, contingent upon the overarching modeling objectives. Because of the physically based foundation of RIM2D, these findings should also be transferable beyond the scope of the presented flood example. Conducting additional sensitivity analyses in future studies on pluvial (Khosh Bin Ghomash et al., 2024b) and coastal (Wu et al., 2010) floods could be valuable, as the interactions between flooding, surface roughness, and terrain resolution may differ from those observed in fluvial floods. For future applications, it is also advisable to explore the integration of advanced data assimilation techniques, such as real-time remote sensing data, to continuously refine model parameters and improve forecasting accuracy. By combining these strategies with the lessons learned from this study, modelers can better tailor the RIM2D model to diverse environments, thereby enhancing its predictive capabilities and supporting more effective flood risk management across various regions.

*Code availability.* RIM2D is available at https://git.gfz-potsdam.de/hydro/rfm/rim2d (last access: 08 October 2024). RIM2D is open-source for scientific use under the EUPL1.2 license. Access is granted upon request. The simulations were performed with version 0.2.

*Data availability.* The land cover raster, which was used to assign roughness values to the simulation domain, is openly accessible at https://www.mundialis.de/en/germany-2020-land-cover-based-on-sentinel-2-data/ (last acess: 08 October 2024).

The 1m DTM has recently been made available at https://gdz.bkg.bund.de/index.php/default/digitale-geodaten/digitale-gelandemodelle/digitales-oberfaechenmodell-dom1.html (last access: 08 October 2024).

Flood extent data were obtained from the UFZ data investigation portal via https://doi.org/10.48758/ufz.14607 (Najafi et al., 2024).

*Author contributions.* SKBG: Conceptualisation, Methodology, Investigation, Simulation, Software, Analysis, Visualisation, Writing; PY, VDN: Conceptualisation, Methodology, Analysis, Visualisation, Writing; HA: RIM2D Software, Conceptualisation, Writing; All authors:
Discussion and Writing.

*Competing interests.* The authors declare no conflicts of interests.

*Acknowledgements.* This research was performed within the frame of the DIRECTED project (https://directedproject.eu/). Funding of the DIRECTED project within the European Union's Horizon Europe – the Framework Programme for Research and Innovation (grant agreement No. 101073978, HORIZON-CL3-2021-DRS-01) is gratefully acknowledged.

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

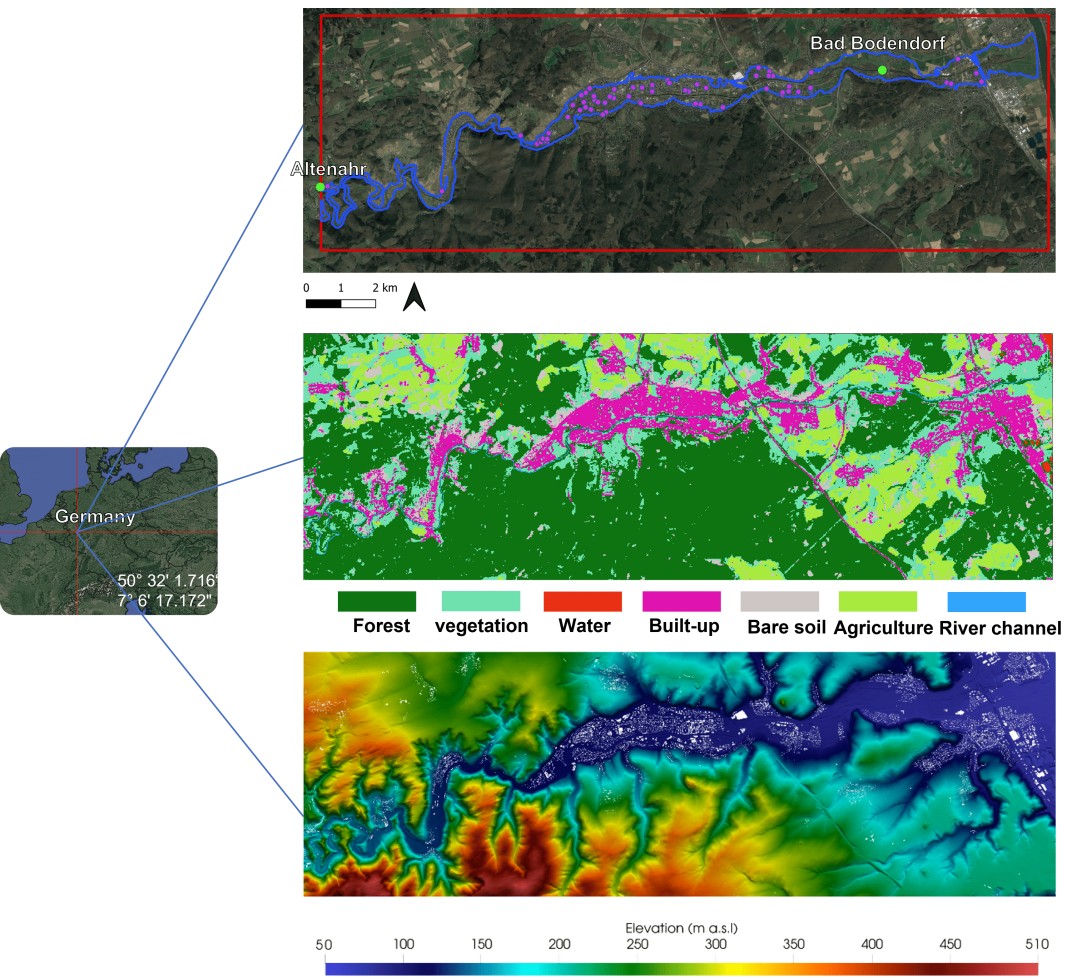

**Figure 1.** The simulation domain's boundary is indicated by the red line in upper figure, the middle figure depicts the land classes and the lower figure illustrates the topography. In the upper figure, the positions of the Altenahr and Bad-Bodendorf gauge stations are denoted by green points. The maximum observed flood extent during the 2021 flood event is represented by the blue line. The purple points represent the locations of the observed water marks during the 2021 flood event. Satellite imagery: © Google Earth 2024

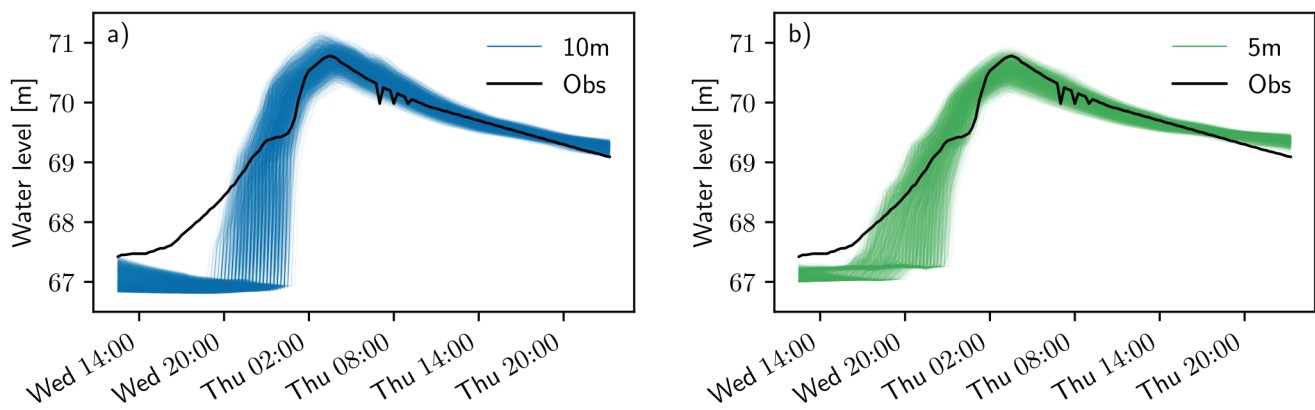

**Figure 2.** Monte Carlo simulation of the in-channel flood dynamics (i.e., water level) at the Bad-Bodendorf gauge station for the (a) dx = 10m and (b) dx = 5m simulations. The black line represents the official reconstructed water level during the 2021 flood event at the gauge.

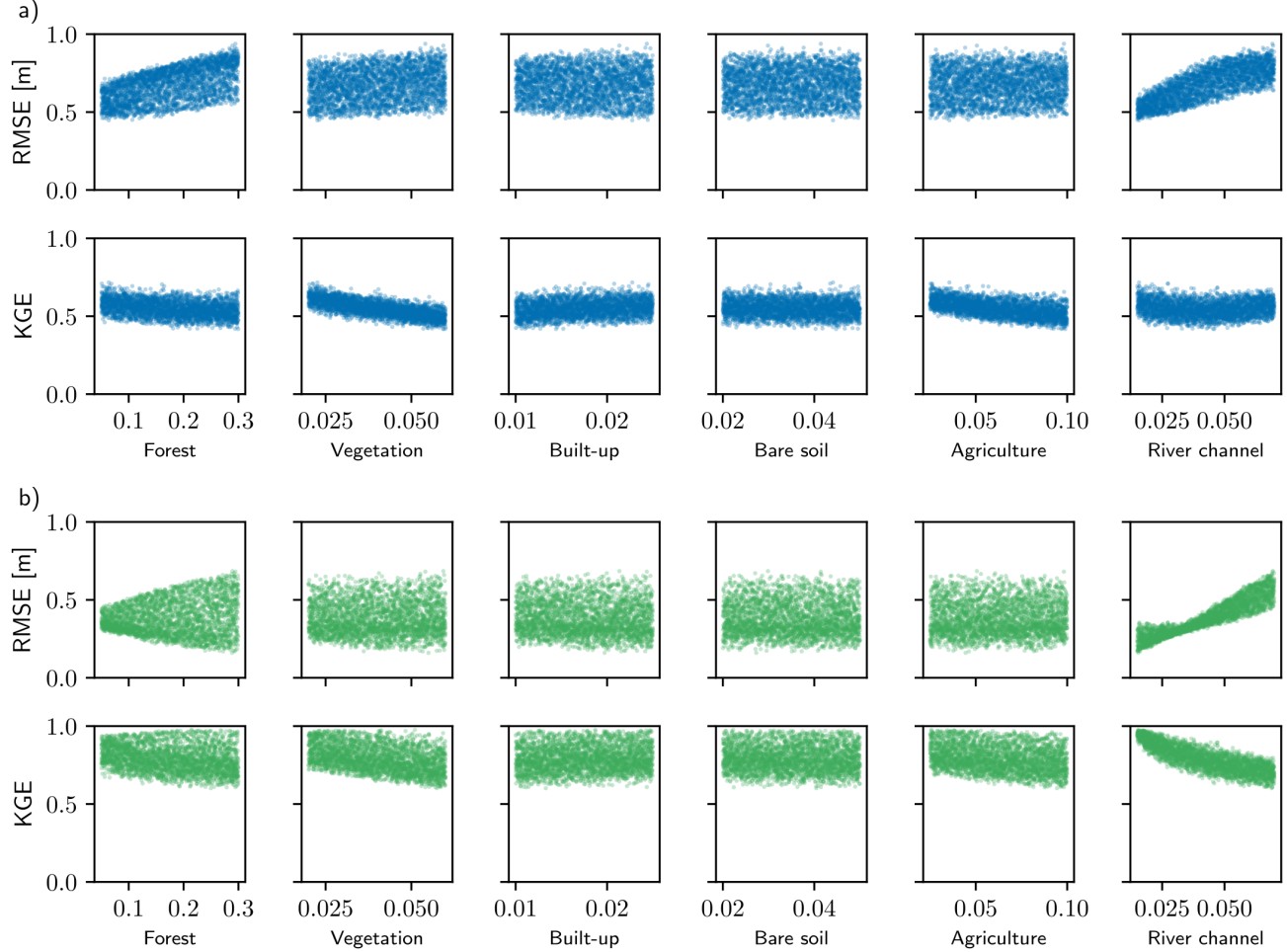

**Figure 3.** Root-Mean-Square Error (RMSE) and Kling-Gupta Efficiency (KGE) for the (a) dx = 10m and (b) dx = 5m simulations calculated for the in-channel flood dynamics at the Bad-Bodendorf gauge. The metrics specific to each land cover category within the simulated area are plotted.

a)

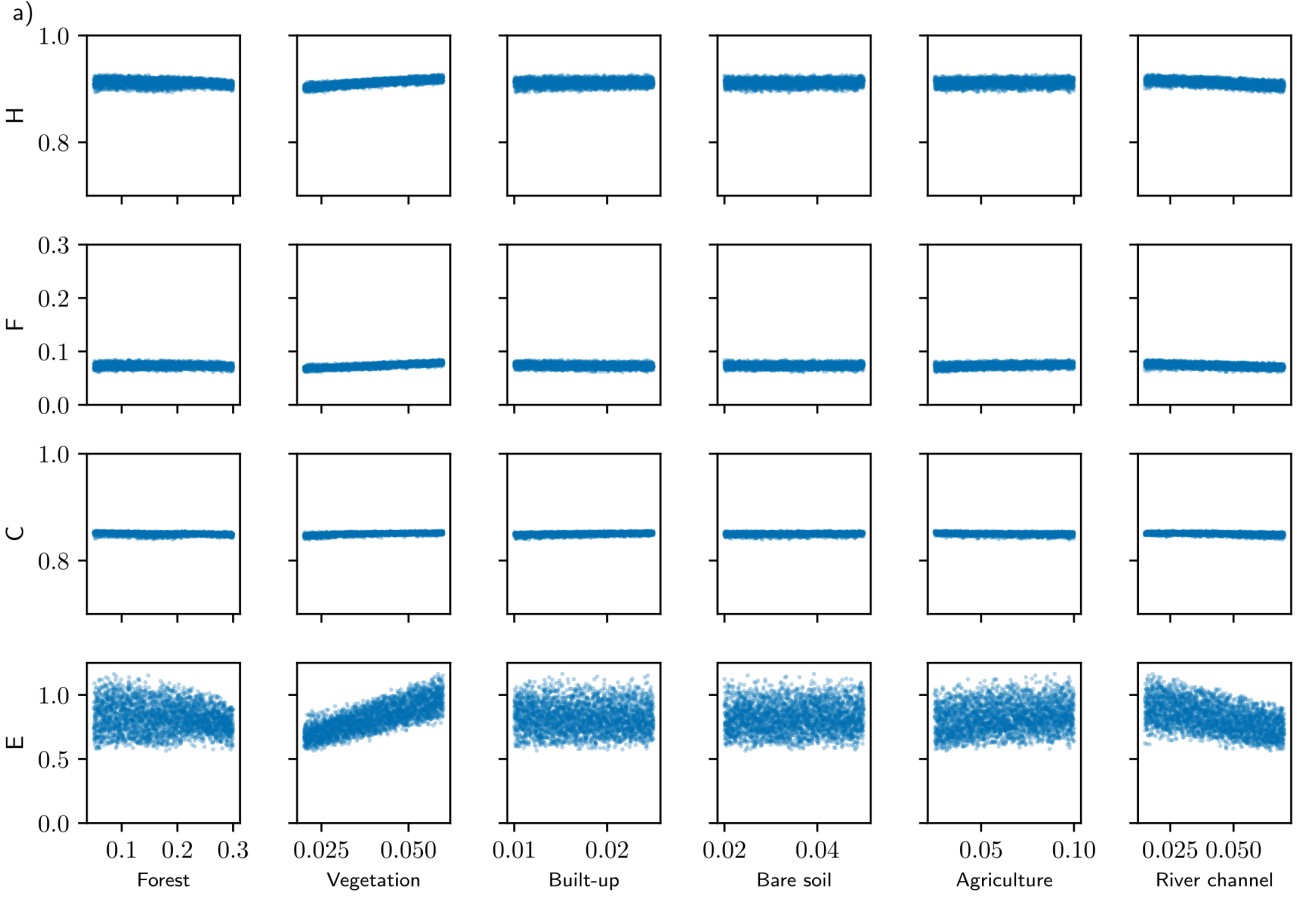

**Figure 4.** Hit rate (H), false alarm ratio (F), Critical Success Index (C) and error bias (E) for the (a) dx = 10m and (b) dx = 5m simulations calculated for the inundated extent. The metrics specific to each land cover category within the simulated area are plotted.

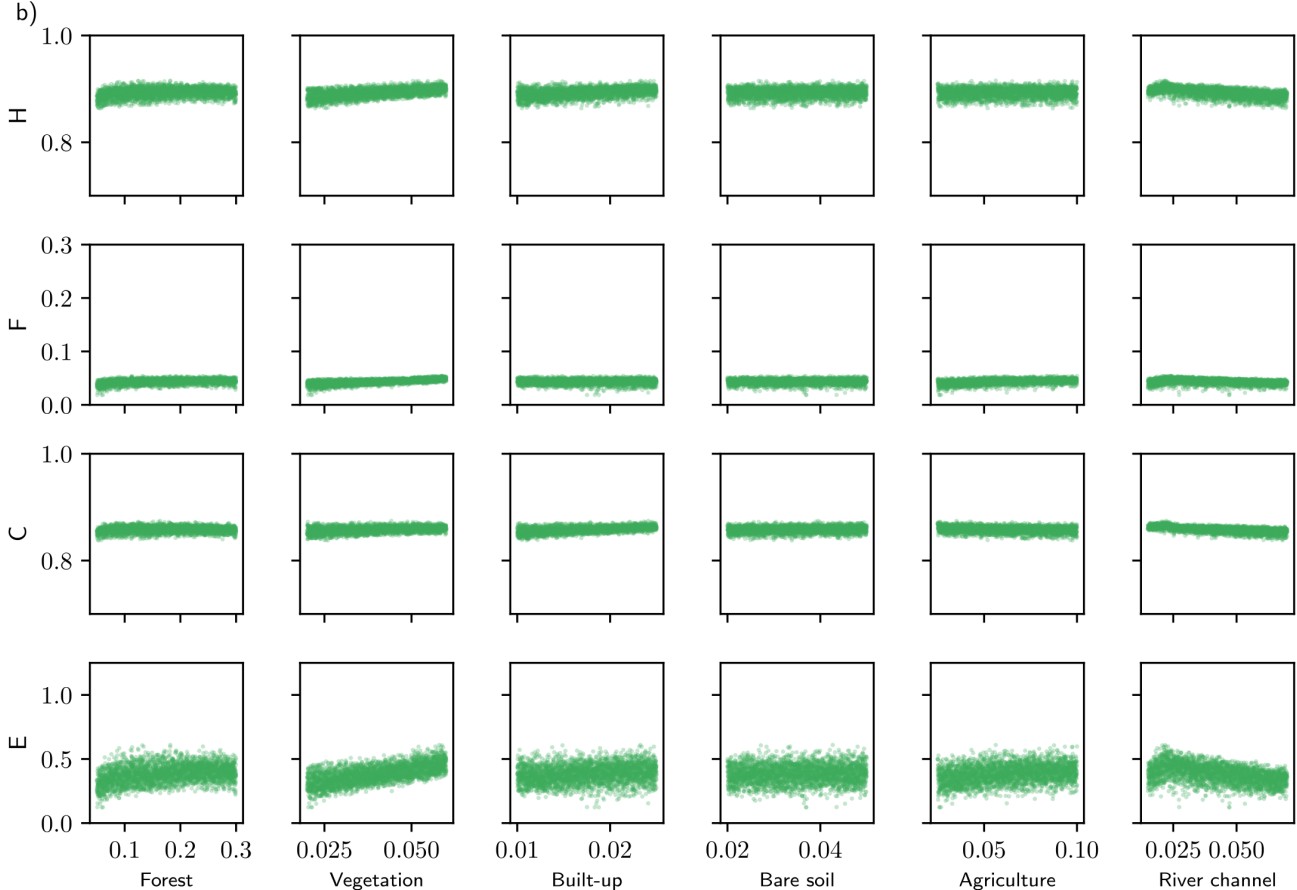

**Figure 4.** Continued.

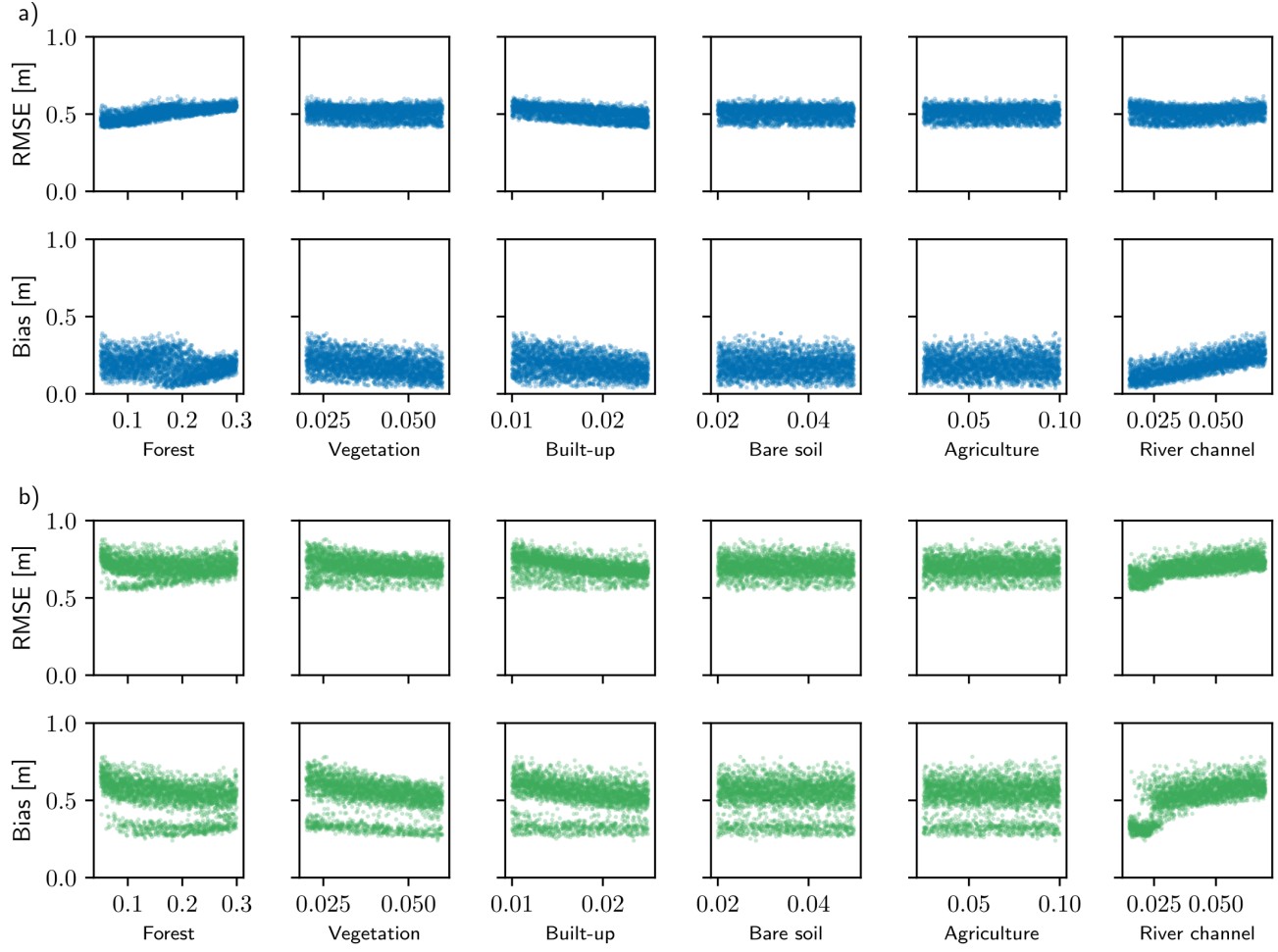

**Figure 5.** Root-Mean-Square Error (RMSE) and bias for the (a) dx = 10m and (b) dx = 5m simulations calculated for the distributed water depths at the high water marks. The metrics specific to each land cover category within the simulated area are plotted.

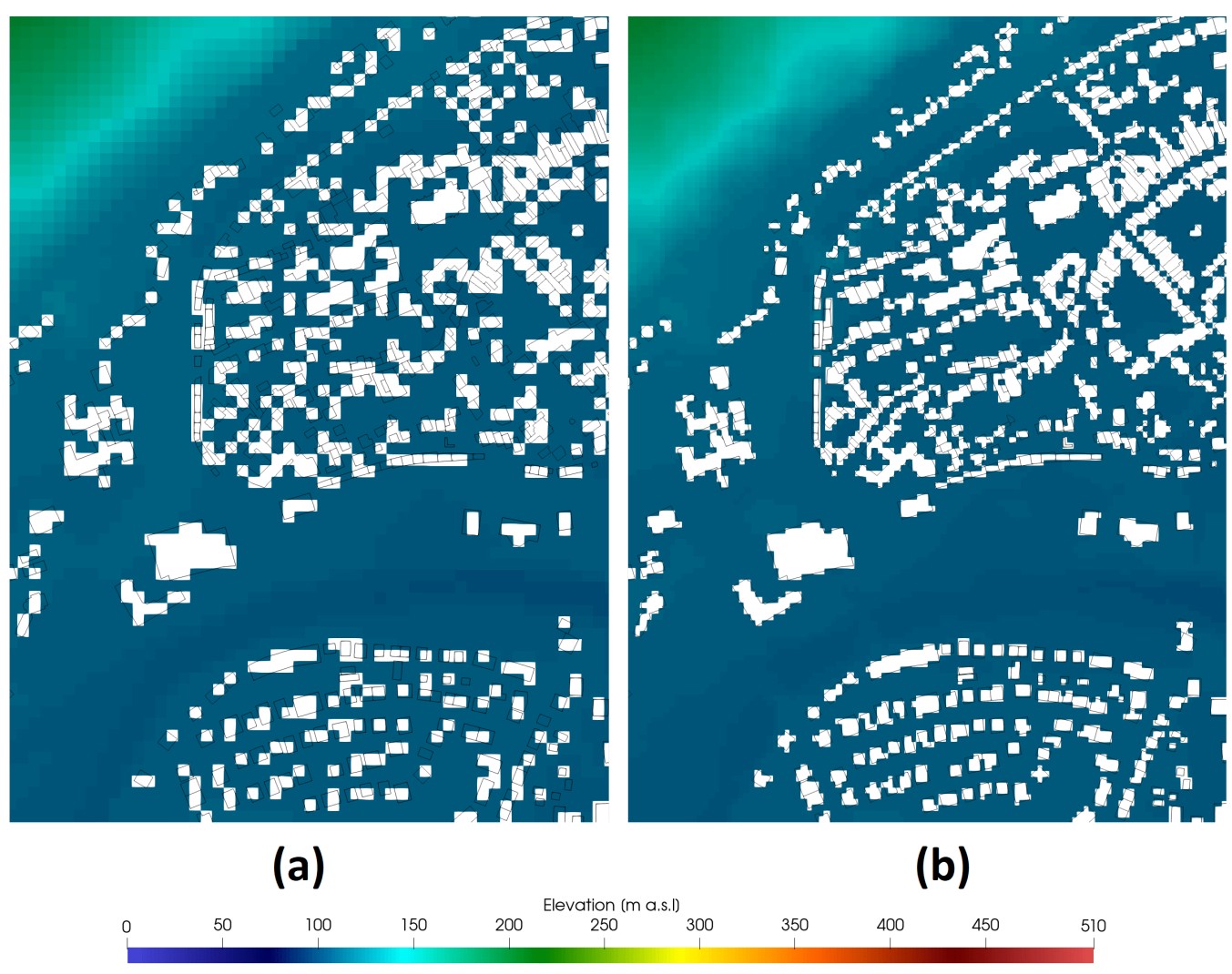

**Figure 6.** OSM footprint of buildings (black lines) and areas of DEM cutouts (white) in the dx = 10 m (a) and 5 m (b) DEMs.

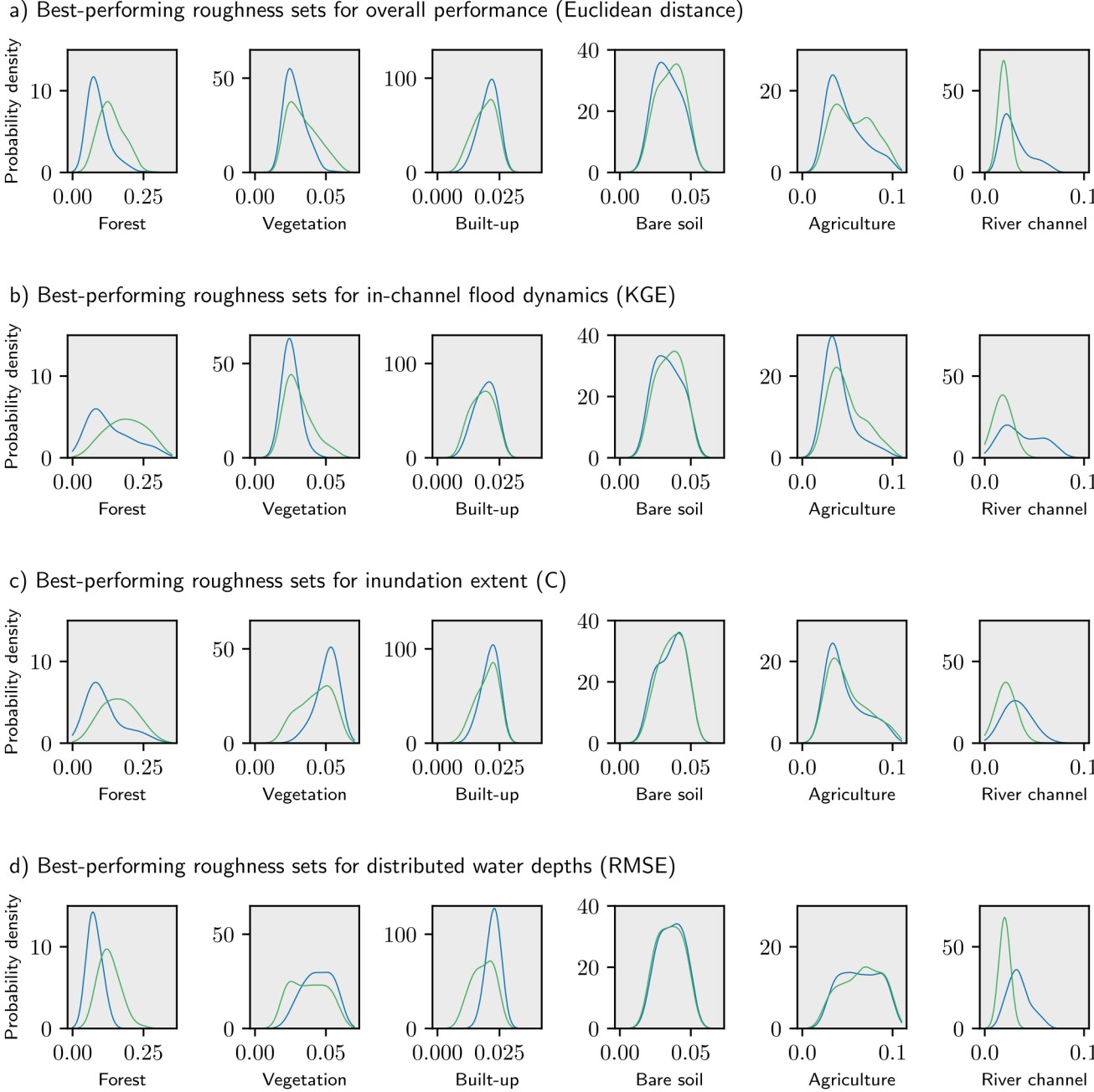

**Figure 7.** Parameter distributions corresponding to the 150 best-performing roughness sets for the dx = 10m (blue lines) and dx = 5m (green lines) simulations according to (a) overall performance (i.e., Euclidean distance), (b) in-channel flood dynamics (i.e., KGE), (c) inundation extent (i.e., $C$) and (d) distributed water depths (i.e., RMSE for distributed water depths). The metrics specific to each land cover category within the simulated area are plotted. The parameter distributions were obtained using a kernel density estimation with varying bandwidths.

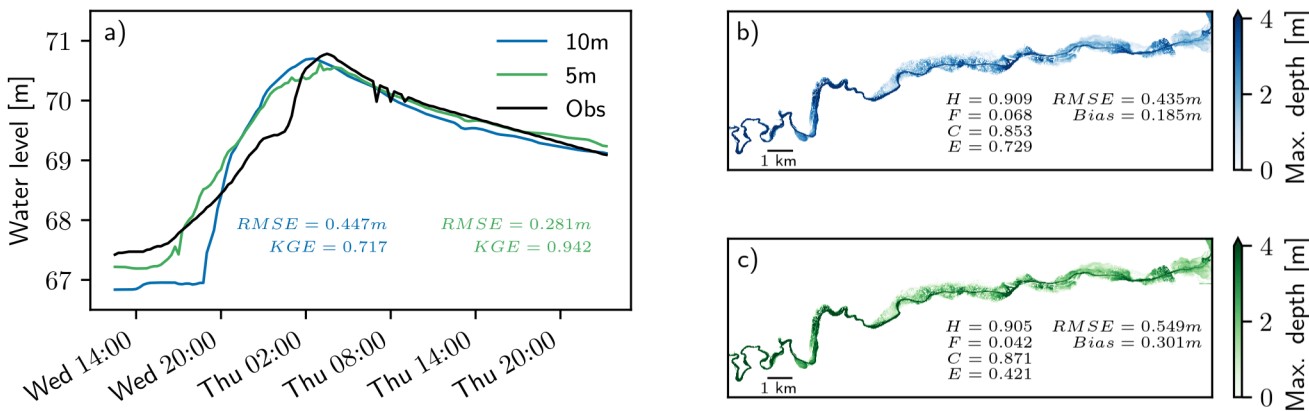

**Figure 8.** Simulations corresponding to the roughness set with the best overall performance (i.e., lowest Euclidean distance) both for dx = 10m (in blue) and dx = 5m (in green). (a) Simulated water levels at the Bad-Bodendorf gauge station. (b) Simulated inundation extent and water depth for the dx = 10m simulation. (c) Simulated inundation extent and water depth for the dx = 5m simulation.

**Table 1.** Land cover categories within the simulation domain, along with the lower and upper bounds of Manning roughness values employed in the simulations. The coverage percentages of these categories across the entire simulation domain and within the observed 2021 flood extent are also presented in the last two columns.

| Land cover category | Lower bound | Upper bound | Coverage percentage [%] | Coverage percentage of the observed flood extent [%] |
|---|---|---|---|---|
| Forest | 0.050 | 0.300 | 52.17 | 14.94 |
| Vegetation | 0.020 | 0.060 | 18.82 | 35.05 |
| Built-up/Sealed areas | 0.010 | 0.025 | 11.37 | 36.33 |
| Bare soil | 0.020 | 0.050 | 4.82 | 2.68 |
| Agriculture | 0.025 | 0.100 | 11.86 | 5.55 |
| River channel | 0.015 | 0.070 | 0.44 | 5.43 |
| Water bodies | - | - | 0.52 | 0.03 |

**Table 2.** Optimal RIM2D roughness ranges estimated as the 5-95 percentile ranges corresponding to the 150 best-performing roughness sets according to overall performance (i.e., Euclidean distance) for the dx = 10m and dx = 5m simulations.

| Land cover category | dx = 10m | | dx = 5m | |
|---|---|---|---|---|
| | **Lower bound** | **Upper bound** | **Lower bound** | **Upper bound** |
| Forest | 0.053 | 0.164 | 0.076 | 0.211 |
| Vegetation | 0.020 | 0.041 | 0.021 | 0.053 |
| Built-up/Sealed areas | 0.015 | 0.025 | 0.011 | 0.024 |
| Bare soil | 0.021 | 0.048 | 0.022 | 0.048 |
| Agriculture | 0.026 | 0.091 | 0.027 | 0.094 |
| River channel | 0.016 | 0.059 | 0.016 | 0.025 |