# Peer review of "Monte-Carlo based sensitivity analysis of the RIM2D hydrodynamic model for the 2021 flood event in Western Germany"

_Natural Hazards and Earth System Sciences, 2024_

## Author Response (AR1)

**Reply to Reviewer 1**

Dear RC1, thank you for taking the time to review our manuscript and provide your comments. please find our replies inlined below.

*1- The study presents the inundation extent, but could you please clarify the water depth associated with this?*

**REPLY:** In this manuscript, we present and compare not only the simulated maximum inundation extents with the observed flood extent but also the corresponding maximum water depths at 65 locations within the domain, where residents reported water levels at their houses after the flood (high water marks = maximum water depths at the building). Figure 5 presents the comparison between the simulated and reported water depths at these locations. Additionally as examples, Figure 8 presents the simulated maximum flood extent and water depths for the roughness set with the best overall performance at resolutions of dx = 10 m (Fig. 8b) and dx = 5 m (Fig. 8c). However to make it more clear in the text that we have also considered water depths in our analysis, we have added a few sentences at the end of Section 3.2.

*2- It's not apparent what the y-axis settings on Figures 3, 4, and 5 reflect. Could you provide further specifics?*

**REPLY:** Figures 3, 4, and 5 display various indicators used to compare model results with observations. Specifically, Figure 3 presents the comparison of simulated in-channel water levels at the Bad Bodendorf gauge, while Figures 4 and 5 show the simulated maximum inundation extent and maximum water depths at 65 locations, respectively. The y-axis in Figure 3 shows the Root-Mean-Square Error (RMSE) and Kling-Gupta Efficiency (KGE) indicators. Figure 4's y-axis displays the Hit rate (H), false alarm ratio (F), Critical Success Index (C), and error bias (E). In Figure 5, the y-axis presents the RMSE and bias indicators. These indicators are explained in Section 3.3, also referenced in the figure captions, and further discussed in the Results and Discussion section of the manuscript.

*3- The study uses a DEM resolution of 1 meter but simulates at 5 meters (5.6 million cells) and 10 meters (1.4 million cells). Typically, the study indicates that finer grids produce more accurate results. Could the authors explain the rationale behind choosing these specific grid sizes for the simulation?*

**REPLY:** Thank you for highlighting this. We, too, would have valued to also have the calibration study including simulations at higher resolutions; however, computational cost was a significant limiting factor. On a single GPU device, the runtimes for resolutions of dx = 5m and dx = 10m are roughly 28.8 and 5.8 minutes, respectively, which makes it feasible to run a large number of simulations (3,000 in our case) for calibration. However, at the DEM's native resolution (dx = 1m), the domain would contain 139 million cells, and at dx = 2m, it would still have 34.7 million cells. The runtimes for these setups are approximately 5.3 hours for dx = 2m and 25.9 hours for dx = 1m. These extended runtimes, combined with our server capacity, make it impractical to run such a large number of simulations for our calibration needs.

Furthermore, \cite{KhoshSWE2024} demonstrated that resolutions of dx = 5 and 10 meters sufficiently capture critical flood dynamics in the Ahr Valley, providing accurate flood extent simulations without requiring finer detail. Also, for a river reach spanning 25 km, using even higher resolutions yields minimal additional value for flood management purposes, as the marginal increase in detail does not enhance the practical usefulness of the results for decision-makers. Instead, such fine resolutions substantially increase computational demands, limiting efficiency without contributing meaningfully to actionable insights. Therefore, the 5- to 10-meter resolution range provides an optimal balance for the presented flood example, delivering reliable flood modeling accuracy while keeping computational requirements manageable.

We have now added text to the manuscript to explain this and include the runtime information.

Khosh Bin Ghomash, S., Apel, H., and Caviedes-Voullième, D.: Are 2D shallow-water solvers fast enough for early flood warning? A comparative assessment on the 2021 Ahr valley flood event, Nat. Hazards Earth Syst. Sci., 24, 2857–2874, https://doi.org/10.5194/nhess-24-2857-2024, 2024.

*4- The study considers Manning's roughness but does not seem to account for uncertainties related to soil cover, slope, and vegetation type, all of which influence floodplain dynamics. How does the RIM2D model account for these factors, and what uncertainties remain?*

**REPLY:** Thank you for highlighting this. However, soil cover and vegetation type are typically considered in hydrological analyses, transforming rainfall into runoff. In a hydraulic study like the presented one, the effect of soils and vegetation on surface water hydraulics is implicitly considered in the roughness calibration, where the roughness classes are based on land use types. However, it has to be noted that in the simulated extreme flash flood event with very high water depths, soil cover and vegetation generally have a very limited impact on the flood dynamics (other than the differentiation between forest and non-forest). The roughness calibration is in this case compensating the effect of the pressure gradient in the water column on the flow (i.e. the missing 3rd dimension in the model), rather than compensating for different impacts of low vegetation types on flow dynamics. Regarding slope, as it is a feature of the area's topography, any uncertainty is tied to the digital elevation model (DEM) used. To minimize this uncertainty, we rely on the highest resolution DEM available for the area, which was obtained through Lidar mapping by the federal state agency.

*5- What criteria were used to optimize the data in Table 2? It would be helpful if the study compared its results with previous work and explained any differences or similarities to strengthen the findings.*

**REPLY:** Table 2 presents the roughness ranges for the 150 top-performing parameter sets based on their overall performance, which was evaluated using the Euclidean distance (ED, as defined in Eq. 1 in the manuscript). The ED serves as a comprehensive metric, incorporating simulated in-channel water levels at the Bad-Bodendorf gauge, the simulated inundation extent, and the simulated distributed water depths across the 65 watermark locations. Thanks for poiting this out, we have now added sentences throughout the manuscript comparing our results to findings from other studies.

*6- Based on the Monte Carlo-based sensitivity analysis of the RIM2D model, what factors could be improved to mitigate future flood events?*

**REPLY:** Thank you for your comment. We have expanded the conclusion of the manuscript to emphasize how the findings of this study can contribute to mitigating future flood events. It has to be noted, however, that the flood simulation is only a tool for understanding the flood dyanmics. This alone does not mitigate flood. A mitigation of the flood impacts can only be achieved by flood manegement and adaption plans, for which the presented simulations and tool is a core tool.

*7- What elements and features should be considered or enhanced to guarantee satisfactory results for further implementations of the RIM2D hydrodynamic model in other areas?*

**REPLY:** This comment has been addressed through the additions made to the conclusion section in response to the earlier comment.

**Reply to Reviewer 2**

Dear RC2, thank you for taking the time to review our manuscript and provide your comments. please find our replies inlined below.

*1- The native resolution of the lidar DEM is 1m (line 97). Given that the model is so sensitive to DEM resolution, I think it would be interesting to add simulations at dx < 5m. A lot of the computational effort is invested into the Monte Carlo analysis, but I think runs at finer resolution could be equally or even more interesting.*

**REPLY:** Thank you for pointing this out and also as replied to the other reviewer, we would have liked to include higher resolution simulations in our calibration study, but computational cost was a major constraint. On a GPU cluster, the runtimes for dx = 5m and dx = 10m are approximately 28.8 and 5.8 minutes, respectively, allowing us to conduct a large number of simulations (3,000 in our case) for calibration. At the DEM's native resolution (dx = 1m), however, the domain contains 139 million cells, and at dx = 2m, it holds 34.7 million cells. The runtimes for these finer resolutions are about 5.3 hours for dx = 2m and 25.9 hours for dx = 1m. Due to these lengthy runtimes and our server capacity, it was not feasible to run large number of simulations for calibration at these resolutions.

Additionally, \cite{KhoshSWE2024} showed that resolutions of dx= 5 and 10 meters capture essential flood dynamics in the Ahr Valley, producing accurate flood extent simulations without the need for finer detail. In addition, for a river reach of approximately 25 km, increasing the resolution beyond this level provides limited added value for flood management, as the additional detail does not significantly improve the practical utility of the results for decision-makers. Instead, higher resolutions greatly increase computational demands, reducing efficiency without adding meaningful insights. Thus, the 5- to 10-meter resolution range offers an ideal balance, ensuring reliable flood modeling accuracy while keeping computational requirements manageable. We have now updated the manuscript to include this explanation and provide runtime information.

Khosh Bin Ghomash, S., Apel, H., and Caviedes-Voullième, D.: Are 2D shallow-water solvers fast enough for early flood warning? A comparative assessment on the 2021 Ahr valley flood event, Nat. Hazards Earth Syst. Sci., 24, 2857–2874, https://doi.org/10.5194/nhess-24-2857-2024, 2024.

*2- If I understand it correctly, the submerged bathymetry of the river was not considered, but the water surface from the lidar model was assumed to be equal to the river bottom elevation. Authors justify this approach by referring to the water level boundaries used in the model. However, wave speed in the river would still be underestimated due to unrealistically shallow depth, which could (partly) explain the delay in the rising limb of the simulated water level time series. Probably the argument here should rather be that the flood event was extreme and thus simulated depths were so high that the ca 1m depth from the lidar model is insignificant. Of course, the magnitude of the error also depends on discharge on the date of lidar acquisition – was that checked? Still, one is left wondering why river bathymetric information was not included – is it unavailable?*

**REPLY:** Thanks for pointing this out. River bathymetric information was not included in our analysis primarily due to the unavailability of such data for the area. Additionally, our aim was to conduct simulations using only the minimal data required (and available) to run these simulations, reflecting the typical conditions in most regions. The advantage of this simplification is that it allows the model to be applied to any river reach without requiring detailed local bathymetric knowledge, enabling a straightforward, semi-automated, and cost-effective implementation. This approach can be applied almost anywhere, as long as a DEM with adequate resolution relative to the river width is available. Also as you noted, this simplification is appropriate for modeling the Ahr Valley flood, which involved extreme flows that significantly exceeded average flow depths (flood flow depths ware about 8-15 times the neglected river bed depth). The impact of the higher river bed elevation in the model as kin reality on the in-channel flow dynamics is small and thus this simplification is justified. We have added some remarks in the manuscript to further clarify this reasoning. Moreover, the elevation along the river channel in the DEM was compared against available real bed elevations (at the site of the gauges). This conformed that the LiDAR DEM was created during normal flow conditions with an average water depth of about 0.8 m (difference DEM to gauge datum).

*3- Line 124ff: The upstream water level boundary is explained in detail. My understanding from further up is that also in the downstream, a water level boundary is used. Please explain how this boundary was implemented – is it the water level in the River Rhine?*

**REPLY:** Thank you for bringing this to our attention. At all boundaries of the simulation domain, we have applied a free outflow condition. The water from the Ahr River flows into the River Rhine and exits through the domain boundaries. To prevent running simulations with empty river conditions, an initial water depth of 1 meter, representing the typical state of the Ahr River, is assigned. The water depth in the Rhine section of the model are higher, representing the deeper channel elevation of the Rhine. Moreover, to ensure more realistic and stable initial river water depths, all simulations include an 18-hour warm-up period, beginning 18 hours before the date shown in Figure 2. We have added some remarks in the manuscript to further clarify this.

*4- The rising limb of the simulated water level time series is not only delayed, but also significantly steeper than what is seen in the reconstructed observations – any explanation? This should be discussed.*

**REPLY:** Thank you for highlighting this issue. We have revised the text in Section 4.1 to address the steeper rising limb in the coarser setup and have included some explanations for this behavior. The main cause for this lies in the particular properties of the hydraulics used and the numerics implemented.

*5- Line 185ff: This issue would actually motivate a coupled 1D-2D modeling approach, using detailed river bathymetry or cross section information, see also review comment 2 above.*

**REPLY:** Indeed, this issue arises when working with DEM resolutions that are too coarse to accurately represent river channels within the domain (in our case at dx = 10 m), a point that our results also highlight. We have revised Section 4.1 in response to the previous comment and have also suggested using 1D-2D coupled approaches, incorporating detailed river bathymetry or cross-section information, as a suitable alternative for future studies when dealing with such coarse DEM resolutions. It has to be noted, though, that the coupling of the

1D and 2D domain is also a source of uncertainty, which can impact on the simulation quality.

*6- Line 207ff: This seems to indicate that the simulation is not "grid-convergent", i.e. results and identified model parameters depend on the chosen grid resolution. This would motivate additional runs with finer grid resolutions as pointed out in comment 1 above. I think it is an important finding of this study that optimal Manning numbers are grid-size dependent, but ideally, they should become grid-size independent for a grid convergent simulation. One is left wondering why the full resolution of the lidar DEM was not exploited to further analyze this issue.*

**REPLY:** This was addressed in our response to the first point. And yes, with this kind of simlpified hydraulic model the roughness parameterization is generally resolution and model setup dependend, because the roughness values are not "real" roughness, but rather effective roughness values compensating for model simplification and resolution. See e.g. Fabio et al. (2010) elaborating on this.

Fabio, P., Aronica, G.T., and Apel, H. (2010). Towards automatic calibration of 2-D flood propagation models. Hydrol. Earth Syst. Sci. 14(6), 911-924. doi: https://doi.org/10.5194/hess-14-911-2010.

*7- Section 4.2: My assumption is that CSI and other metrics were calculated by comparing maximum simulated flood extent with the observations? Or was the comparison done for the exact time of image acquisition? It is also unclear which technique was used to get "observed" flood extent. Please clarify.*

**REPLY:** Thank you for bringing this to our attention. Indeed, the comparison is between the simulated maximum inundation extent and the observed extent. Flood extent was derived from airborne mapping of flood extent after the event, supplemented and corrected with field observations of inundation extent (detemined by debris lines). Meaning that the observation maps the maximum flood extent. We have now added text to the manuscript to clarify this point. Additionally, we have included the observed flood extent data in the data availability statement of the manuscript.

*8-Line 340ff: I think this is an important point. Validating such complex models with just one time series is insufficient. It is important to use spatio-temporally distributed water surface elevation datasets from satellite missions such as SWOT and ICESat-2 and/or drones (e.g. https://doi.org/10.1016/j.rse.2019.111487, https://doi.org/10.3390/rs12071171)*

**REPLY:** Thank you, we have now emphasized this point in the conclusion.

*Details:*

*L18: Replace "often" with "frequent"*

*L187: Delete "be"*

*L224: Re-arrange wording*

*L 235 ff: Reword "Manning roughness values" instead of "roughness Manning values".*

**REPLY:** Thanks for highlighting these error. They have now been corrected.